# Blockchain-Enabled Traceability in the Rice Supply Chain: Insights from the TRACE-RICE Project

**DOI:** 10.3390/foods14213711

**Published:** 2025-10-30

**Authors:** Carlota Gonçalves, João Fernandes, Carla Brites

**Affiliations:** 1National Institute for Agricultural and Veterinary Research (INIAV), I.P., Av. da República, 2780-157 Oeiras, Portugal; carlotamateusgoncalves@outlook.pt (C.G.); joao.fernandes@iniav.pt (J.F.); 2GREEN-IT Bioresources for Sustainability, ITQB NOVA, Av. da República, 2780-157 Oeiras, Portugal

**Keywords:** blockchain, rice supply chain, traceability

## Abstract

Agri-food supply chains, particularly in the rice sector, face persistent challenges in transparency, quality control, and sustainability due to their complexity and fragmentation. Blockchain technology provides a promising solution by ensuring secure, immutable, and verifiable records of production and supply chain activities, supporting both consumer trust and compliance with the EU Common Agricultural Policy (CAP). This study reports on the TRACE-RICE Mediterranean pilot project, which developed a blockchain-enabled traceability system for rice production in Portugal. A Rice Field Data Recording App, built with ArcGIS Survey123, digitized agronomic and compliance records from Integrated Production systems and linked them to blockchain-verified QR codes on consumer packaging. The pilot conducted during the 2023 harvest demonstrated the potential to enhance data consistency and streamline field recording processes, thereby improving transparency in farming practices. A total of 174 QR code interactions, primarily from Lisbon, revealed consumer engagement patterns valuable for future business analysis. The scaling phase during the 2024 harvest confirmed the system’s adaptability to different varieties and production contexts, positioning blockchain as a replicable model for sustainable and competitive rice supply chains.

## 1. Introduction

Agri-food supply chains—particularly in the rice sector—are inherently complex, involving a diverse network of actors, from smallholder farmers and primary processors to traders, manufacturers, distributors, retailers, and consumers. This complexity often limits traceability and creates vulnerabilities in transparency, quality assurance, and sustainability. Blockchain technology offers a transformative pathway to address these challenges by ensuring data integrity, enabling real-time supply chain visibility, and supporting compliance with sustainability frameworks. Beyond operational benefits, blockchain can contribute to the United Nations Sustainable Development Goals (SDGs) by tackling structural issues such as inequality, poverty, human rights violations, and environmental degradation [1].

Blockchain’s decentralized architecture, immutability, and capacity for automating transactions through smart contracts offer robust safeguards for supply chain operations [1]. Takahashi and Lakhani [2] highlight its potential to reduce security risks, while Wenhua et al. [3] demonstrate that carefully designed encryption, authentication, and smart contract protocols can ensure high levels of data integrity and protection. Nevertheless, Kumar and Mallick [4] warn that in developing countries, limited infrastructure and technological expertise can exacerbate cybersecurity vulnerabilities.

Although initial investment costs can be substantial, the potential benefits of blockchain adoption in agri-food systems are well recognized [5,6]. High implementation expenses may hinder uptake, but the technology’s capacity to reduce food loss, strengthen quality control, and enhance consumer trust through verifiable transparency makes it a strategically valuable innovation [7]. Nonetheless, further research is required to align blockchain’s technical capabilities with policy frameworks, market dynamics, and the digital readiness of agri-food stakeholders.

In the European Union, the Common Agricultural Policy (CAP) 2023–2027 places strong emphasis on product differentiation based on quality, transparency of food origin, and explicit mapping of supply chain actors. Achieving these objectives increasingly depends on the adoption of digital solutions such as blockchain to support Farm-to-Fork traceability [8]. A key policy shift is the requirement that farmers applying for subsidies under the Integrated Production certification scheme [9] maintain a digital farm book—replacing paper-based records to enhance data reuse and reporting efficiency.

Given the scale of rice production and its central role in global diets, the establishment of robust traceability systems is essential. Such systems not only safeguard food safety but also promote sustainable agricultural practices and respond to growing consumer expectations for transparency [10,11]. Within this policy and technological landscape, the TRACE-RICE pilot project in Portugal developed a mobile application to capture field-level data from rice cultivated under Integrated Production systems. The application generates a unique QR code that links to dynamic labels containing real-time, verified origin information. By connecting farm-level data collection with consumer-facing transparency, TRACE-RICE provides reliable information on product origin and farming practices, while demonstrating how blockchain-based tools can reconfigure rice supply chains, align with CAP objectives, and strengthen both sustainability and consumer trust.

This article reviews the design and implementation of the TRACE-RICE App for rice field data recording, its integration with consumer-facing QR codes, and assesses the barriers and opportunities for scaling blockchain-based traceability systems across diverse rice supply chain contexts.

## 2. Blockchain Technology in Agri-Food Traceability

Blockchain is a distributed ledger technology that enables secure, immutable, and transparent recording of transactions among multiple stakeholders without requiring centralized control. In agri-food supply chains, its potential lies in providing verifiable proof of origin, ensuring product authenticity, and creating auditable records for safety, quality, and sustainability claims [12]. A major barrier to blockchain adoption is businesses’ reluctance to share sensitive information, driven by low trust among supply chain partners [13]. This hesitation undermines the collaborative foundation essential for blockchain’s effectiveness. One solution lies in applying data encryption techniques to restrict access and safeguard information. Furthermore, adopting a distributed file system across the network can ensure continuous file availability, even during network failures [14]. By recording each transaction as a cryptographically linked “block” in a sequential “chain,” blockchain creates an incorruptible audit trail accessible to all authorized parties, adding an extra layer of control and transparency that reinforces quality assurance processes and supports compliance with voluntary certification schemes and industry standards [15].

### 2.1. Key Technical Features

The architecture of blockchain systems can be public, private, or consortium-based, depending on the governance model and access rights. Public blockchains allow full transparency but may suffer from scalability issues and high transaction costs. Private blockchains restrict access to vetted participants, providing faster processing and better control over sensitive data. Consortium blockchains, which are commonly used in agri-food applications, combine these advantages by allowing a group of organizations to maintain the ledger collaboratively [16].

Core features relevant to agri-food traceability include decentralization, which eliminates dependence on a single authority and reduces the risk of data tampering; immutability, meaning once data is recorded, it cannot be altered without consensus from network participants; transparency, which enables stakeholders to access the same verified information in real time and fosters trust; and smart contracts, which are self-executing protocols that trigger actions such as payments or alerts when predefined conditions are met.

### 2.2. Application in Agri-Food Supply Chains

Blockchain integration in agri-food systems can address common traceability challenges by ensuring origin verification through linking crop production records with processing and distribution data, thereby ensuring authenticity and helping combat fraud [17]. It can improve food safety by rapidly identifying contamination sources, which reduces recall times and associated economic losses. Additionally, blockchain supports sustainability reporting by capturing environmental metrics such as carbon footprint or water usage, which aids certification schemes and compliance reporting. Furthermore, transparent quality records enable market differentiation by allowing premium pricing for products that meet specific consumer preferences. However, successful adoption of blockchain requires interoperability with existing information systems, robust mechanisms for verifying data inputs, and training for stakeholders.

### 2.3. Relevance to the Rice Sector

Rice is one of the most traded food commodities globally, with a supply chain that is both geographically dispersed and structurally fragmented. In the Mediterranean context, the sector includes small-scale farmers, cooperatives, milling facilities, exporters, and retailers operating under varying levels of digital readiness. Issues such as inconsistent quality control, origin mislabelling, and insufficient sustainability reporting make rice an ideal candidate for blockchain-enabled traceability solutions.

TRACE-RICE deployed blockchain to link farm-level data (captured via mobile applications) with consumer-facing labels through QR codes. This allowed consumers to verify not only geographical origin but also production practices and quality parameters.

## 3. The TRACE-RICE Case Study: Design and Pilot Implementation

The TRACE-RICE project was conceived to design, develop, and pilot a blockchain-enabled traceability system tailored to the Mediterranean rice supply chain. Its primary objective was to strengthen transparency, quality control, and sustainability metrics while ensuring that the system could operate seamlessly from paddy field to consumer. The pilot implementation in Portugal focused on integrating digital field data with blockchain-based verification, thereby offering consumers transparent and verifiable information while enhancing the competitiveness of rice producers.

The traceability system was developed through the creation of a Rice Field Data Recording App, integrated with consumer-facing QR codes and supported by a blockchain infrastructure, complemented with digital tools to capture, validate, and disseminate rice production data. Field agents and farmers used a mobile application to record agronomic practices, harvest dates, and input usage. These data were uploaded to a cloud platform, verified, and subsequently stored in the blockchain ledger. A consortium blockchain, managed collaboratively by stakeholders, guaranteed immutability through hashed transaction records and controlled access. For consumers, packaging was equipped with QR codes that linked to blockchain-verified webpages, offering detailed provenance and quality information.

### 3.1. Development of the Rice Field Data Recording App Using ArcGIS Survey123

A central component of TRACE-RICE was the Rice Field Data Recording App, developed using ArcGIS Survey123, a flexible ESRI software tool for digital data collection. The App was designed to align with the specific requirements of rice under the Integrated Production certification scheme, while ensuring regulatory compliance, thus offering a comprehensive and practical solution for rice producers. The decision to adopt ESRI technology was guided by its well-established ecosystem, including ArcGIS Pro, ArcGIS Online, and Survey123 Connect, which collectively provide robust resources for geospatial data management and long-term accessibility for agricultural stakeholders.

The App development followed a structured process beginning with the creation of a project map in ArcGIS Pro. This was subsequently converted into a web map through ArcGIS Online to ensure accessibility for end-users. Integration with the Portuguese Agriculture and Fisheries Financing Institute (IFAP) database provided geographic data specific to rice plots in Portugal, which was then linked to the Survey123 App through Survey123 Connect. Pre-loaded data further streamlined data entry and editing, enabling farmers to interact with the application more efficiently.

#### 3.1.1. Survey Structure

The App survey template was adapted from an existing paper-based format [18] and optimized to capture comprehensive and structured field data. It was divided into five main sections, each addressing a critical aspect of rice farming details and operations:(i)Identification of Farm Plots: Included geographic details and ownership records of each plot.(ii)Field or Crop Cycle Records: Captured phenological state evolution, pest and disease monitoring, crop management practices, plant protection treatments and fertilization records.(iii)Mechanical Operations: Logged mechanical interventions in the field.(iv)Harvest Records: Documented harvesting details, including dates and yields.(v)Observations and Photographs: Allowed for additional notes and visual documentation of the rice fields.

#### 3.1.2. Features for Efficiency and Accessibility

The App included several advanced features to streamline operations and ensure usability:

Pre-loaded Data: To facilitate user interaction and improve efficiency, pre-loaded information was integrated into the App. It included the European Catalog of Rice Varieties [19], providing a standardized reference for rice types, a list of Plant Protection, providing a standardized reference for rice varieties, a list of Plant Protection Treatments [20] approved for rice cultivation and a predefined list of Mechanical Operations commonly associated with rice farming.

Multilingual Capability: To enhance accessibility, the App in Portuguese was translated into English, Spanish and Arabic, targeting key rice producer countries such as Spain and Egypt. Using ArcGIS Survey123’s multilingual functionality, a single Excel form supported all languages without requiring separate templates.

Integration of Portuguese Rice Plots Database: A critical aspect of the App was its ability to interface with farmers’ registrations with the Portuguese payment authority IFAP, enabling access to detailed geographic information about rice plots in Portugal. IFAP provides publicly available geographic data services for mainland Portugal [21], including geographic data layers such as crop type, land use, and plot information. Since no specific dataset for rice cultivation is available, a Portuguese Rice Plot Database was created using ArcGIS. This was achieved by filtering and merging relevant data from three existing IFAP layers: plot locations, crop types, and land use, divided by 18 districts of Portugal.

The integration process of rice plot data, illustrated in Figure 1 (Schematic Workflow), involved four main steps.

The process comprises four main steps:(i)Layer Selection and Field IdentificationThree spatial data layers from the IFAP database were used: Crop Types Layer (“Culturas”)—includes fields: OBJECT_ID, CUL_ID, CUL_CODIGO, OSA_ID, CT_PORTUGUÊS; Land Use Layer (“Ocupações de Solo”)—includes fields: OBJECT_ID, OSA_ID, VDO_DESCRI, PAR_ID; Plot Locations Layer (“Parcelas”)—includes fields: OBJECT_ID, PAR_ID, PAR_NUM.(ii)Filtering Relevant DataIn the Crop Types Layer, records were filtered using the field CT_PORTUGUÊS for the values “Arroz”, “Elemento Linear Arroz”, “Elemento Linear em Orizicultura”; in the Land Use Layer, records were filtered using VDO_DESCRI for the values “ELP Orizicultura”, “Orizicultura”, “Culturas Temporárias”. These filters ensured that the dataset focused exclusively on rice cultivation.(iii)Data Joining and IntegrationThe Crop Types and Land Use layers were joined using the common field OSA_ID, creating a dataset combining crop type and land use information. The resulting dataset was then joined with the Plot Locations Layer through the common field PAR_ID. This integration generated a unified dataset linking all relevant geospatial and agronomic information for rice plots across Portugal.(iv)Optimization and Index CreationTo improve database performance and streamline data access, an index was created in ArcGIS Pro, which optimized join operations and accelerated queries.

The final georeferenced rice plot database enhanced operational workflows by simplifying access to verified plot data and supporting efficient integration with the TRACE-RICE data collection App.

The App digitized data collection on geographic location, crop phenology, plant protection, fertilization, mechanical operations, and harvest details—following Integrated Production principles and CAP regulations. It generated chronological, detailed reports that provide reliable documentation for compliance verification and audit purposes across the rice production chain.

The Spanish version benefited from alignment with digital farm management systems developed by the Ministerio de Agricultura, Pesca y Alimentación [22,23,24], while the Arabic translation was prepared in collaboration with the University of Alexandria to ensure cultural and technical accuracy.

#### 3.1.3. Survey App Template Published as Open Access

The Rice Field Data Recording App was made available as an open-access resource on the ArcGIS Survey123 platform, allowing users to adapt its Excel-based framework to their local needs. This open approach encourages scalability and knowledge transfer beyond the Portuguese pilot.

### 3.2. Digital Field Data Recording App During the Pilot Phase

The 2023 rice harvest marked the pilot phase for evaluating the Rice Field Data Recording App developed under the TRACE-RICE project. Conducted in collaboration with Ernesto Morgado S.A. (EM), a leading rice miller, the pilot aimed to validate the App functionality and implement a digital traceability system linking detailed field data to consumers through QR codes. A rice plot in the Mondego region, owned by EM, was selected for this purpose.

During the research phase of the pilot, several critical parameters that the App needed to address were identified. These included ensuring data integrity through integrated backup procedures, guaranteeing accessibility on mobile devices via intuitive interfaces, and securing compatibility with both iOS and Android operating systems. Ease of use was also considered essential, with workflows designed to accommodate both GIS specialists and non-specialist users. The App further provided field access to relevant reference materials, such as plots reported to payment authorities and high-resolution satellite imagery, while optimizing the overall flow of information to improve usability and efficiency.

### 3.3. Blockchain Architecture During the Pilot Phase

In this pilot, the blockchain solution integrated field-level data collected through the ArcGIS Survey123 App into a database and blockchain infrastructure developed within the Microsoft Power Platform environment. The system used Power Apps for user interaction, Power Automate for workflow automation, and Dataverse as the secure, permissioned ledger supporting data storage and traceability.

A dedicated connector automated data capture from the rice milling industry’s (Ernesto Morgado S.A.) operational 4D database via a SOAP (Simple Object Access Protocol) web service, covering all activities from paddy purchase to packaged rice sales. Each transaction generated a unique hash stored in the blockchain ledger, ensuring tamper-proof traceability and data integrity.

For the consumer interface, QR codes were dynamically generated for each rice lot, containing unique identifiers linked to the corresponding blockchain entries. These codes provided real-time transparency through web or mobile access and functioned as both a traceability tool and a marketing asset. The system configuration and data flow, from field data capture to consumer transparency, are illustrated in Figure 2, which presents the blockchain-enabled rice traceability architecture implemented in the TRACE-RICE pilot project.

Comprehensive reports from this phase are provided in Appendix A.

### 3.4. Performance Evaluation

To assess the effectiveness and scalability of the TRACE-RICE App, several quantitative and qualitative benchmarks were established.

Usability was evaluated based on intuitive navigation and user feedback from participating farmers. Efficiency was measured through average data upload latency, blockchain write/read times, and the reduction in data entry errors compared to paper-based records. Traceability was verified through the linkage of field data to consumer-facing QR codes. During the pilot, 174 QR code scans were recorded, primarily from Lisbon, indicating consumer engagement and geographical reach.

Additionally, user adoption rates among farmers confirmed the App’s operational feasibility and adaptability to different production contexts. These results demonstrate measurable improvements in data accuracy, operational efficiency, and compliance with Integrated Production principles, establishing a solid foundation for broader implementation across Mediterranean rice supply chains.

### 3.5. Pilot Scaling up in the 2024 Harvest

During the 2024 rice harvest, the pilot initiative was expanded to include the newly released Portuguese Caravela rice variety, with monitoring extended to three additional plots in the Tejo region. Detailed field data were collected from these plots, and QR codes were generated for each monitored parcel, further reinforcing the model’s scalability and adaptability across different production environments. At this stage, the blockchain infrastructure was implemented solely for field data recording, focusing on the secure and transparent registration of agronomic practices. A hybrid storage model was adopted: only hashed transaction metadata and verification timestamps were recorded on-chain, ensuring data integrity and immutability, while the raw field data were stored off-chain on INIAV secure servers. This approach minimized blockchain storage requirements, improved performance, and safeguarded sensitive information in compliance with GDPR.

While interoperability with external databases such as IFAP and DGADR has not yet been implemented, it is envisioned as a future development to enable real-time synchronization of parcel data, compliance verification, and enhanced scalability under the Common Agricultural Policy (CAP) framework.

In parallel, this phase expanded the project’s analytical scope beyond traceability to explore the relationship between field conditions and rice quality.

This approach demonstrates the technical feasibility of implementing blockchain-based traceability at the field level, establishing a structured basis for a scalable, interoperable, and transparent digital framework within the rice supply chain.

## 4. Recommendations for Blockchain Implementation

The integration of blockchain technology into the rice supply chain holds transformative potential for enhancing transparency, security, and traceability throughout the entire value chain. However, successful adoption requires overcoming technical, economic, and regulatory challenges while actively engaging all stakeholders.

### 4.1. Cryptography and Access Control

To ensure data integrity, confidentiality, and GDPR compliance, the TRACE-RICE pilot implemented a layered security approach. All field and operational data stored in the blockchain were hashed using SHA-256, providing tamper-evident records. Data transmission between the ArcGIS Survey123 App, the industrial partner 4D operational database (via SOAP web service), and the blockchain infrastructure was encrypted using a combination of symmetric and asymmetric cryptography. Access to the blockchain and the Microsoft Power Platform interface was managed through role-based permissions and Azure Active Directory authentication, ensuring that only authorized personnel could view or modify records. Private–public key pairs were centrally managed, assigned to organizational roles rather than individual users, simplifying key management while maintaining strong authentication and auditability. This layered approach provides a robust framework for secure data capture, storage, and retrieval, reinforcing both operational integrity and regulatory compliance across the rice supply chain.

### 4.2. Reducing Costs and Technical Barriers

The primary barriers to blockchain adoption in agriculture are high costs and technological complexity. While cheaper consensus protocols may lower costs, they risk compromising core blockchain attributes such as decentralization, security, scalability, and transparency [25]. Despite initial investment hurdles, blockchain solutions have demonstrated reduced operational costs over time compared to centralized databases [5]. As the technology matures, costs are expected to decrease, improving accessibility for smaller producers.

Lightweight blockchain solutions integrated with mobile tools—such as the Rice Field Data Recording App—offer a practical path forward. Utilizing decentralized and edge-computing techniques reduces processing demands while maintaining data integrity. Compatibility with existing agricultural databases (e.g., IFAP and DGADR in Portugal) ensures a smooth transition for producers familiar with digital tools. Addressing these barriers is critical to enabling scalable blockchain adoption, especially in resource-limited contexts.

### 4.3. Policy Recommendations to Ease Regulatory Hurdles

A supportive regulatory environment is essential to facilitate blockchain adoption. Current limitations include the absence of standardized public-key infrastructures, interoperable platforms, and harmonized regulations [26,27]. Coordinated efforts with agricultural organizations to harmonize standards at regional and international levels would simplify stakeholder operations and promote interoperability between public and private ledger systems [28,29]

Policy incentives such as subsidies could help offset initial blockchain implementation costs and encourage early adoption. Additionally, integrating blockchain initiatives into broader sustainability frameworks—such as the EU’s Common Agricultural Policy (CAP) and the United Nations Sustainable Development Goals (SDGs)—would align blockchain use with global environmental and social objectives, increasing stakeholder motivation [1].

### 4.4. Stakeholder Engagement and Buy-In Strategies

A critical factor in the pilot’s success was the active engagement of stakeholders across the rice supply chain. Training sessions and continuous feedback loops helped refine both the App’s usability and the underlying data protocols, while addressing barriers such as digital literacy among smallholders and ensuring compliance with data privacy regulations. Demonstrating the neutrality and security of the blockchain network fostered trust and encouraged adoption.

Successful blockchain traceability depends on stakeholder trust and engagement—producers, millers, regulators, and consumers alike. Targeted training and outreach, emphasizing tangible benefits such as improved food safety, market differentiation, and operational efficiency, strengthen buy-in. Involving stakeholders early in system design ensures solutions meet real-world needs, while transparent communication and data-driven evidence, such as QR code analytics, build credibility. Public awareness efforts highlighting blockchain’s role in food quality and provenance further stimulate consumer demand for traceability solutions [30].

To enhance accessibility, two video tutorials and demonstrations of the Rice Field Data Recording App and QR code integration have been published on the official TRACE-RICE YouTube channel, available in Portuguese and English. These resources guide farmers, policymakers, and the public on the App’s functionality and benefits, playing a key role in promoting adoption.

### 4.5. Scaling and Replicating Success

The TRACE-RICE pilot demonstrated that blockchain-enabled traceability can serve as a transferable model for other agri-food chains facing similar challenges in data integrity, regulatory compliance, and consumer transparency. To maximize its broader impact, the Rice Field Data Recording App and blockchain architecture have been conceptualized into a TRACE-RICE Agri-Blockchain Scalability Framework. The framework consists of three interlinked layers designed to ensure interoperability, resilience, and adaptability across diverse agricultural systems (Figure 3):The Data Layer establishes standardized data schemas and metadata for interoperability with national agricultural databases (e.g., IFAP, DGADR in Portugal), enabling hybrid on-chain/off-chain data storage through secure APIs.The Blockchain Layer ensures data immutability, auditability, and consortium-based governance via a permissioned blockchain (Microsoft Power Platform environment) operating under a Proof of Authority (PoA) consensus for energy-efficient validation.The Application Layer delivers customizable interfaces for field data capture (via ArcGIS Survey123), smart contract functionalities (e.g., compliance verification, payment release), and consumer transparency tools (QR codes).

The framework follows a phased deployment strategy: starting with smallholder pilots for iterative validation and progressively scaling to multi-node, multi-actor systems across the agri-food sector. Feedback loops between layers ensure continuous optimization, maintaining cost-effectiveness and user engagement as data volume and complexity increase.

By publishing the Rice Field Data Recording App on the ArcGIS Survey123 platform, TRACE-RICE establishes a replicable, service-oriented model integrating geospatial data collection, blockchain-based validation, and transparent communication with consumers. The TRACE-RICE Agri-Blockchain Scalability Framework, thus, provides a generalizable blueprint for the sustainable adoption of blockchain solutions across global agri-food systems—extending well beyond rice production.

### 4.6. Comparative Benchmarking of TRACE-RICE with Other Blockchain Agri-Food Systems

To contextualize the TRACE RICE system within the broader agri-food blockchain landscape, a comparative benchmarking analysis was performed against two established frameworks—iFoodDS [31] and RiceChain. The iFoodDS platform delivers real-time traceability, quality, and food-safety capabilities across supplier, processor, distributor and retail networks. In contrast, RiceChain presents a blockchain-based framework for rice supply chains grounded in smart contracts, focusing on traceability from farm to consumer. In comparison, TRACE-RICE adopts a consortium governance model integrated with ArcGIS-based field data collection and European regulatory databases, offering a policy-aligned, scalable and cost-effective framework tailored to Mediterranean rice systems. A detailed comparison of their defining characteristics is presented in Table 1.

### 4.7. Lessons Learned and Future Recommendations

The TRACE-RICE pilot implementation generated practical insights into the application of blockchain technology within Mediterranean rice supply chains, identifying both the potential benefits and the technical and organizational challenges that require resolution to achieve scalable and sustainable adoption. The results indicate that, although blockchain offers a robust framework for immutable data recording, its overall effectiveness is primarily determined by the accuracy and timeliness of data input.

To mitigate input errors and ensure reliable records, the integration of automated data capture tools, such as IoT-based environmental sensors, yield monitors, and grain quality analyzers, was identified as a strategic next step. Although not yet fully implemented in this pilot, these technologies are planned for future deployment to enable automated data ingestion and validation through smart contracts, reducing discrepancies and reinforcing confidence in the recorded information. Furthermore, implementing multi-layered validation protocols involving cross-checks between actors reduced discrepancies and increased confidence in the system.

Connectivity issues in rural areas emerged as a significant constraint, underscoring the need for offline data capture capabilities with subsequent synchronization when connectivity is restored. Designing lightweight and user-friendly mobile applications that function well under limited network conditions is paramount for wider adoption, especially among smallholders with limited digital literacy.

Data privacy and governance were other critical considerations. Ensuring compliance with GDPR and local regulations while maintaining transparency required carefully balanced access controls and permissioned blockchain networks. Clear agreements on data ownership and sharing rights helped avoid conflicts and built trust among participants.

Looking ahead, future developments should focus on further integration of traceability systems with existing farm management and certification platforms to streamline workflows and reduce duplication of efforts. Enhancing interoperability with other supply chain actors beyond the rice sector could open avenues for cross-commodity sustainability certifications and market expansion. Moreover, exploring the use of advanced analytics and AI-driven insights based on blockchain data could empower stakeholders with predictive capabilities and decision support.

To maximize impact, scaling up the TRACE-RICE model will require strong institutional support and public–private partnerships to fund infrastructure investments and capacity-building programs. Encouraging policy frameworks that recognize and incentivize blockchain traceability can accelerate adoption and reinforce competitiveness on the international stage.

One of the most important lessons was the critical role of stakeholder engagement and training. Active involvement of farmers, cooperatives, and other supply chain actors proved essential to foster trust and encourage consistent data entry. Continuous education on the use and benefits of the system helped overcome initial skepticism and improved data quality over time. Additionally, engaging local agricultural extension services and technical support teams enhanced user adoption and system troubleshooting.

The TRACE-RICE case study indicates that blockchain-enabled traceability represents a viable approach for enhancing transparency, quality control, and sustainability in Mediterranean rice value chains. With further technical development and active stakeholder engagement, these digital tools have the potential to support the transition toward more resilient and reliable agri-food systems.

## 5. Conclusions

TRACE-RICE developed the Rice Field Data Recording App, a digital tool designed to enhance transparency, traceability, and consumer confidence in rice production. The pilot phase demonstrated the system’s functionality in improving data collection efficiency, supporting regulatory compliance, and generating operational insights across the supply chain. Results highlight dual benefits: enhanced traceability through QR codes and valuable analytics that support data-driven decision-making for producers and retailers.

Scalability was confirmed during the 2024 harvest, where expanded monitoring of the Portuguese Caravela rice variety illustrated the App’s adaptability to diverse agricultural contexts. This iterative approach underscores its potential for broader deployment.

The integration of blockchain technology marks a paradigm shift for rice supply chains. Beyond ensuring transparency and food safety, blockchain fosters sustainability by linking agricultural practices to global objectives such as the Sustainable Development Goals (SDGs). Its alignment with regulatory frameworks, including the EU’s Common Agricultural Policy (CAP), further enhances compliance and market differentiation.

As blockchain matures, with lower costs and greater accessibility, smaller producers will be better positioned to adopt these tools. Supported by policy incentives, stakeholder engagement, and scalable implementation models, blockchain is poised to become a cornerstone of agricultural traceability systems.

TRACE-RICE establishes a reference model for innovation in rice supply chain management and offers a replicable framework for integrating blockchain technology into agri-food systems. By fostering collaboration, promoting sustainability, and enhancing transparency, the initiative illustrates the potential of blockchain to improve the quality, traceability, and accountability of food systems along the entire value chain.

## Figures and Tables

**Figure 1 foods-14-03711-f001:**
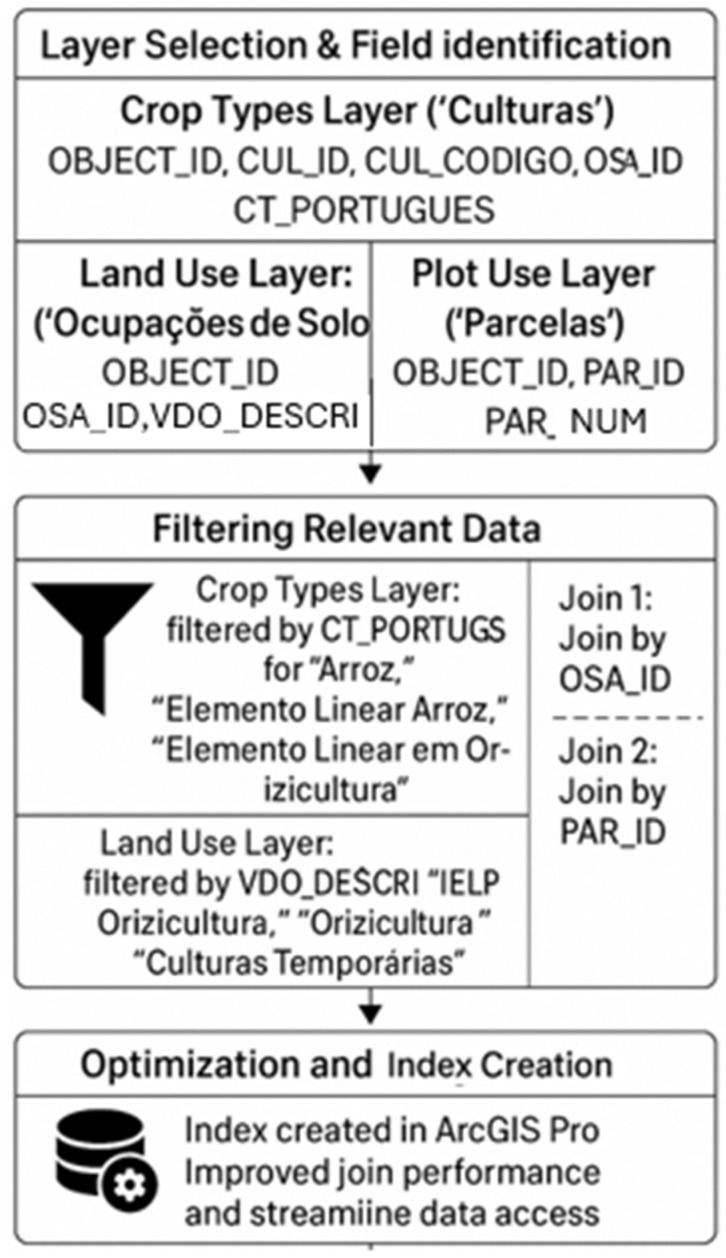
Schematic Workflow of Rice Plot Data Integration and Optimization Using ArcGIS Pro.

**Figure 2 foods-14-03711-f002:**
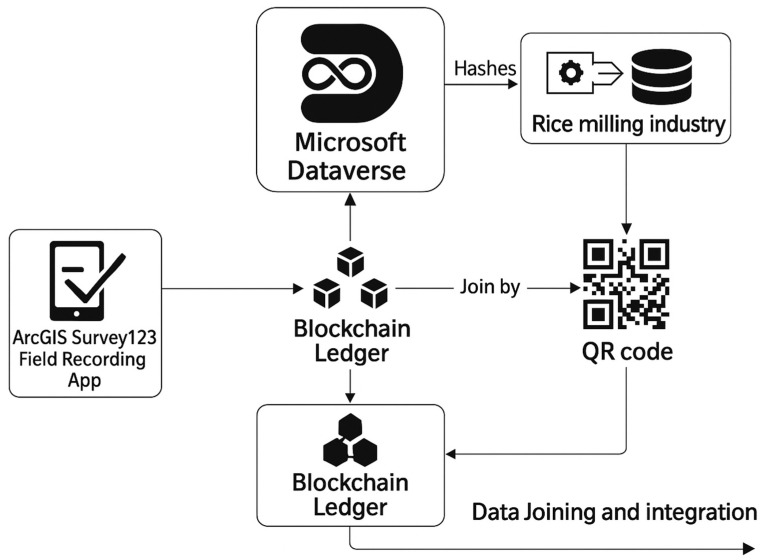
System Architecture for Blockchain-Enabled Rice Traceability in the TRACE-RICE Pilot Project.

**Figure 3 foods-14-03711-f003:**
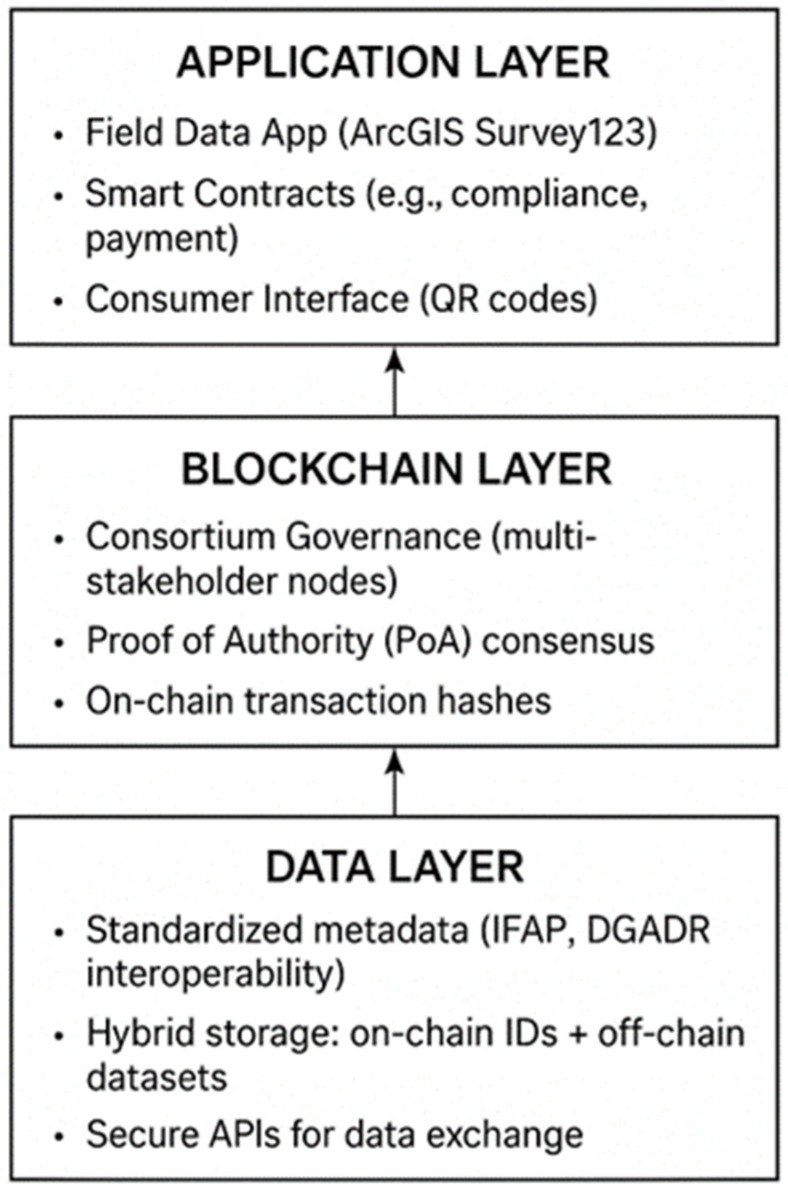
The TRACE-RICE Agri-Blockchain Scalability Framework integrates three functional layers: Data, Blockchain, and Application to ensure modularity, scalability, and interoperability in agri-food traceability systems.

**Table 1 foods-14-03711-t001:** Comparative Features of Blockchain Systems in Agri-Food Supply Chains.

Feature	TRACE-RICE (Mediterranean)	iFoodDS	RiceChain (Asia)
Blockchain Type	Consortium-based, tailored for agrifood stakeholders	Traceability software platform (cloud-based); not primarily described as a full blockchain network	Public/private (Ethereum smart-contract framework)
Governance Model	Multi-stakeholder consortium (farmers, cooperatives, regulators, researchers)	Enterprise network solution for food service, retail and upstream suppliers	Smart-contract network among rice value-chain stakeholders
Data Integration	Field-level data capture via ArcGIS Survey123; interoperable with EU regulatory DBs	Real-time visibility across supply chain; supplier data link, compliance with FSMA 204	Smart contracts record transactions from farm to consumer
Scalability/Focus	Designed for Mediterranean rice chains, cost-efficient for small/medium producers	Enterprise-scale solutions for large networks of suppliers and processors	Focused on rice supply chain traceability and consumer feedback
Policy Alignment	Aligned with EU CAP 2023–2027, Farm to Fork, regulation frameworks	Aligned with US FSMA 204 traceability requirements	Oriented toward agricultural product safety, traceability and smart-contract-based monitoring

## Data Availability

The TRACE-RICE survey template is published in the ArcGIS Survey123 platform and is accessible to licensed ESRI customers. The raw data collected from farmers using this survey are securely stored on the INIAV institutional server and are available under controlled access due to privacy and data protection regulations. Aggregated and anonymized information generated from Survey123 reports is publicly accessible through the reading of the QR code printed on the rice packaging.

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
