# Peer review of "Blockchain-Enabled Traceability in the Rice Supply Chain: Insights from the TRACE-RICE Project"

_foods, 2025, doi:10.3390/foods14213711_

Round 1
Reviewer 1 Report
Comments and Suggestions for Authors
This study developed and piloted a blockchain-based rice supply chain traceability system (TRACE-RICE). Field data was digitally collected using the ArcGIS Survey123 mobile application and linked to blockchain verification QR codes on consumer packaging. The following issues remain to be addressed:
1. Key technical details are vaguely described, compromising the study's reproducibility. Examples include the specific blockchain implementation and the QR code generation and association mechanisms.
2. The results section lacks empirical data, presenting assertions over evidence. Claims such as “improved data accuracy, operational efficiency” and “provided valuable business insights” are made without any quantitative or qualitative data to support these conclusions.
3. Some expressions, while not hindering comprehension, lack precision. Review the entire text and refine all overly absolute, colloquial, or imprecise sentences to ensure objective and rigorous language.
4. Multiple inconsistencies and errors exist in the reference formatting. Each citation must be meticulously reviewed and revised to strictly comply with MDPI journal reference style requirements.
5. The “Data Availability Statement” must explicitly specify data access permissions, clarifying whether data is publicly available, under controlled access, or restricted to authors only.
6. Adding additional figures and tables is recommended to enhance data visualization and readability.
Reviewer 2 Report
Comments and Suggestions for Authors
As a postdoctoral scholar working in AI and cybersecurity with experience in blockchain-based systems for data integrity and traceability, I find this paper — “Blockchain-Enabled Traceability in the Rice Supply Chain: Insights from the TRACE-RICE Project” — to be a well-structured applied case study. It effectively bridges blockchain technology with agri-food systems, demonstrating a real-world implementation through the TRACE-RICE App. The manuscript clearly conveys technical design choices (e.g., ArcGIS Survey123 integration, consortium blockchain governance) and aligns the project with EU policy frameworks such as CAP and the SDGs. However, while the practical implementation is well-documented, the scientific contribution and novelty beyond system integration remain somewhat limited. The work would benefit from deeper technical analysis (e.g., blockchain architecture performance, data security protocols, or IoT integration) and clearer articulation of how this pilot advances the state of research beyond existing literature.
- The paper identifies the use of a consortium blockchain but lacks clarity on the specific platform (e.g., Hyperledger Fabric, Ethereum Private Network). Details such as consensus mechanism, block validation time, transaction throughput, and node management are essential for reproducibility and to assess scalability and energy efficiency.
- While the study mentions hashed transaction records and GDPR compliance, it does not elaborate on encryption methods, access control layers, or how private–public key management was operationalized. Given the sensitivity of agricultural data, a subsection explaining the cryptographic standards (SHA-256, RSA, etc.) and authentication flows would strengthen technical rigor.
- The evaluation primarily focuses on usability and traceability from a qualitative standpoint. The authors should include quantitative benchmarks such as data upload latency, blockchain write/read time, user adoption rate, and QR code response analytics to validate performance and efficiency.
- Although the 2024 expansion is mentioned, there is little discussion on the blockchain’s scalability under increased data volume or multi-node operation. A technical discussion comparing on-chain vs. off-chain storage strategies and interoperability with external APIs (e.g., IFAP, DGADR systems) would enhance understanding of system resilience.
- The paper briefly references IoT in the recommendations, but its integration is not experimentally validated. Demonstrating a prototype smart contract (e.g., automated compliance verification or payment release) or IoT-based automatic data ingestion would provide a stronger link between theory and practice.
- Some sentences are overly long and could be split for clarity. For example, in Section 3.1.2 (“The process included four steps…”), restructuring complex enumerations into bullet-style lists or schematic diagrams would enhance readability and reduce redundancy.
- While the implementation is strong, the manuscript does not introduce new blockchain frameworks or optimization techniques. The contribution is mainly applied. The authors could enhance impact by formulating a generalized model or framework for agri-blockchain scalability and governance applicable beyond rice production.
- The paper does not benchmark TRACE-RICE against other blockchain-based agri-food systems (e.g., IBM Food Trust, RiceChain). Including a comparative performance or policy alignment analysis would help quantify the system’s unique contributions and situate it within the broader research ecosystem.
Round 2
Reviewer 2 Report
Comments and Suggestions for Authors
The revised version is acceptable.